# Mitigating Unintended Memorization with LoRA in Federated Learning for LLMs

## Abstract

Federated learning (FL) is a popular paradigm for collaborative training which avoids direct data exposure between clients. However, data privacy issues still remain: FL-trained large language models are capable of memorizing and completing phrases and sentences contained in training data when given with their prefixes. Thus, it is possible for adversarial and honest-but-curious clients to recover training data of other participants simply through targeted prompting. In this work, we demonstrate that a popular and simple fine-tuning strategy, low-rank adaptation (LoRA), reduces memorization during FL up to a factor of 10. We study this effect by performing a medical question-answering fine-tuning task and injecting multiple replicas of out-of-distribution sensitive sequences drawn from an external clinical dataset. We observe a reduction in memorization for a wide variety of Llama 2 and 3 models, and find that LoRA can reduce memorization in centralized learning as well. Furthermore, we show that LoRA can be combined with other privacy-preserving techniques such as gradient clipping and Gaussian noising, secure aggregation, and Goldfish loss to further improve record-level privacy while maintaining performance.

## 1 Introduction

Large language models (LLMs) have been shown to achieve state-of-the-art performance over most relevant natural language processing (NLP) tasks (Zhao et al., 2023). There is an emerging and significant interest in fine-tuning LLMs to conduct tasks over specialized domains such as medicine (Thirunavukarasu et al., 2023; Yang et al., 2022) and finance (Wu et al., 2023b; Li et al., 2023). These fields handle inherently sensitive user data, necessitating additional mechanisms to prevent data exposure. A well-studied paradigm for collaboratively training a machine learning (ML) model over a cluster of clients without sharing local data is federated learning (FL) (McMahan et al., 2016; Kairouz et al., 2021).

Although FL respects data sovereignty by allowing training samples to remain decentralized, most FL works do not address the memorization problem: an FL-trained LLM may still memorize client training data. Indeed, memorization is observable in most, if not all, LLMs (Carlini et al., 2019; 2022; 2021), with some work arguing that memorization is required to learn natural speech patterns (Dourish, 2004; Feldman, 2020). While there is a wealth of research focused on preventing data reconstruction (Huang et al., 2021) and improving differential privacy (El Ouadrhiri & Abdelhadi, 2022) within the FL literature, very few have explored the propensity and prevention of FL-trained LLMs to leak training data (Thakkar et al., 2020).

In this work, we demonstrate an intuitive and efficient strategy for reducing memorization during LLM fine-tuning: low-rank adaptation (LoRA) (Hu et al., 2021). In fact, we observe that LoRA fine-tuning mitigates regurgitation of synthetically-injected sensitive data in both the federated and centralized settings. This includes exact token matching (Carlini et al., 2022) and approximate reproduction (Ippolito et al., 2023). As LoRA combines the benefits of reduced computational (Hu et al., 2021), memory (Dettmers et al., 2024), and communication overhead (Liu et al., 2024), its added benefit of preventing memorization makes it an ideal strategy for FL fine-tuning of LLMs.

Our contributions are as follows:

- We discover and demonstrate that LoRA mitigates memorization in federated and centralized learning. This includes exact match rate (repeating training data exactly) and paraphrasing (partial overlap). Compared to full fine-tuning, LoRA can significantly reduce memorization even when sensitive data is replicated and the LLM is prompted with long prefixes of a sequence.

- We comprehensively test models of varying size from the Llama-2 family, Llama-3 family, and Mistral-v0.3 on medical question-answering tasks to simulate a data-sensitive scenario. LoRA effectively reduces memorization while preserving high performance accuracy.

- We experimentally explore how LoRA interacts with other privacy strategies. This includes differential privacy mechanisms such as gradient noising and clipping, Goldfish loss (Hans et al., 2024), and post-training noise injection. We find that LoRA works synergistically with these other approaches.

- We will publicly release our code after the review process.

## 2 RELATED WORK

### 2.1 PRIVACY IN LLMS

Exposure of sensitive data via generative models has been extensively considered in existing literature, though the choice of the privacy evaluation metric continues to evolve.

**Differential privacy.** Classical $(\epsilon, \delta)$-differential privacy (DP) frameworks formally measure the privacy-preserving capacity of an algorithm by analyzing whether the probability of observing an output changes by $\epsilon$ when the underlying database excludes or includes a user record (Dwork et al., 2006). The application of this framework to generative language tasks, in general, has proven complicated due to the rigid definition of a user record (Jayaraman & Evans, 2019). When directly applying DP to prevent sensitive data reconstruction, it has been shown that a non-negligible compromise on privacy is required to maintain performance (Lukas et al., 2023). The conventional technique of adding Gaussian noise onto clipped gradients (Abadi et al., 2016) to boost privacy has also been shown to affect model outputs: the randomness of the noise alone can significantly alter the outputs of two equally-private models (Kulynych et al., 2023). One must consider the context and length of a prompt that goads an LLM into leaking sensitive information (Nissenbaum, 2004; Dourish, 2004) – a condition absent from the DP perspective (Brown et al., 2022).

**Memorization.** The ability of language models (large or otherwise) to regurgitate pieces of their training data is well-documented. However, the question of *how best* to quantify the memorization capacity of an LLM is an active area of research. A seminal work by Carlini et al. introduced "canaries", which are synthetic, out-of-distribution pieces of text injected into training data (such as `"My SSN is XXX-XX-XXXX"`) (Carlini et al., 2019). The approach is computationally expensive, as it requires perplexity comparisons against many thousands of random sequences, and canaries should be inserted anywhere from 1 to 10,000 times to gather a full picture of exposure, thus requiring significant fine-tuning. However, it has found use in production-level studies (Ramaswamy et al., 2020) and adjacent fields such as machine unlearning (Jagielski et al., 2022). An alternative proposal of memorization (Carlini et al., 2022), the completion metric, adopted by our work, measures how often an LLM completes a piece of text taken from the training text when prompted on an initial portion (prefix) of it.

### 2.2 FEDERATED LEARNING

**Privacy in FL.** Federated learning, although initially designed to protect user data (McMahan et al., 2017), did not foresee leakage in the form of regurgitation as its advent preceded the development of high-performing generative language models (Kairouz et al., 2021). Consequently, studies on the memorization capacity of FL-trained LLMs remain limited. An early survey demonstrated that federated averaging (Thakkar et al., 2020) ameliorates unintended memorization, though only for a tiny 1.3M parameter next-word predictor (Hard et al., 2018). However, the authors' observations on the success of non-independent and identically distributed (non-IID) clustering for improved privacy informed our federated training strategy. The addition of the DP Gaussian mechanism was shown to improve canary-based memorization for a production FL setting (Ramaswamy et al., 2020).

Similar to us, Liu et al. (2024) leverage LoRA to conduct efficient fine-tuning. However, this work is exclusively interested in studying performance under varying budgets within the $(\epsilon, \delta)$-DP framework and does not consider memorization under the canary or completion-based framework.

**Medical applications.** Our emphasis on medical datasets is relevant: LLMs have been shown to regurgitate sensitive medical data in Lehman et al. (2021), though their work relies on an older BERT model. Mireshghallah et al. (2022) study the success of membership inference attacks on i2b2, though they also do not use any memorization metrics. Although federated learning has been studied and championed as an ideal paradigm for clinical settings (Xu et al., 2021; Nguyen et al., 2022; Antunes et al., 2022), there is a relative lack of literature in the context of clinical memorization.

## 3 PRELIMINARIES

**LoRA.** To reduce computational and memory requirements when fine-tuning LLMs, Low-Rank Adaptation (LoRA) (Hu et al., 2021) was introduced to drastically reduce the number of trainable parameters while fine-tuning. This is achieved by representing the weight updates $\Delta W$ as the product $\Delta W = BA$ of two low-rank matrices $A$ and $B$. LoRA enables efficient adaptation of LLMs to specific tasks while preserving the generalization capabilities of the underlying model, as gradients often exhibit a low intrinsic dimension (Li et al., 2018; Aghajanyan et al., 2020). Additionally, LoRA offers a notable advantage in an FL scenario by drastically reducing the amount of data exchanged between participants during each round. In our experiments, we achieved a reduction by a factor of 130.

**Federated Learning.** Federated learning (FL) has been widely-studied for deep learning models in cross-silo settings Huang et al. (2022), where a limited number of resource-rich clients, such as organizations or institutions, collaboratively train ML models without sharing their data. In conventional FL, the global objective function of $N$ clients is defined as

$$\min_W F(W) = \sum_{k=1}^{N} p_k f_k(W), \tag{1}$$

where $W$ represents parameters of a model, $\sum_{k=1}^{N} p_k = 1$ and $f_k(W)$ is the local objective function of client $k$. Local training data $\mathcal{D}_k$ between clients often heterogeneous. A common strategy for solving Equation 1 is Federated Averaging (FedAvg) (McMahan et al., 2016). In FedAvg, clients conduct a round $t$ of training and $\theta_{t+1}$ (parameters after round $t$) is updated as the $p_k$-weighted average of the respective $k$ gradients. These gradient weights $p_k$ can be set as $p_k = \frac{|\mathcal{D}_k|}{\sum_{k=1}^{N} |\mathcal{D}_k|}$ to mitigate data size bias, which we use in this work. FL has been recently applied to LLMs Ye et al. (2024); Thakkar et al. (2020); Liu et al. (2024); Ramaswamy et al. (2020) leveraging FedAvg to aggregate locally-trained model updates. In this work, we conduct experiments using LoRA-based fine-tuning and full model fine-tuning for local iterations in FL. Besides reducing communication costs, clients benefit computationally from using LoRA during local training.

**Memorization Definition.** Following previous work (Ippolito et al., 2023; Huang et al., 2024; Hans et al., 2024), we adopt the "extractable memorization" definition of Carlini et al. (2023). Consider a string representable as a concatenation $[p||s]$ where $p$ is a prefix of length $k$ and $s$ is the remainder of the string. We define the string $s$ to be *memorized with $k$ tokens of context* by a language model $f$ if $[p||s]$ is contained in the training data of $f$, and $f$ produces $s$ when prompted with $p$ using greedy decoding. In other words, we consider a string from training data memorized if an LLM can generate it when prompted by a prefix.

## 4 EMPIRICAL EVALUATION

In this section, we study how LoRA affects memorization of out-of-distribution sequences injected into fine-tuning training data. We introduce the experimental setting in Section 4.1 and explain how we quantify memorization in Section 4.2.

We consider conventional centralized learning in Section 4.3, where all training samples are trained on by a single client. We then consider an FL setting in Section 4.4, where training data is split among several clients. Our FL experiments are designed to mimic a medical setting where training

data contains sensitive information at an unknown rate, which is a common scenario as few if not any data anonymization tools can guarantee a complete removal of sensitive data (Langarizadeh et al., 2018). In fact, Heider et al. (2020) measured the accuracy of three off-the-shelf de-identification tools on the i2b2 medical record dataset (Stubbs & Özlem Uzuner, 2015), which our experiments also use, and found that no system could perform a full removal.

## 4.1 EXPERIMENTAL SETUP

All fine-tuning was performed on a single NVIDIA A100 80GB GPU within an HPC cluster. We leveraged HuggingFace's Transformers library (Wolf et al., 2020) to access and fine-tune pre-trained models. The experiments were conducted in a Python 3.11.9 environment, with PyTorch 2.4.0 and CUDA 12.1. Further training details are included in Appendix B.1.

We fine-tune models for domain adaptation to medical question-answering (QA). Despite medical scenarios being extensively promoted by FL applications (Xu et al., 2021; Nguyen et al., 2022; Antunes et al., 2022), and the availability of resources such as de-anonymized sensitive medical datasets (Johnson et al., 2016; Stubbs & Özlem Uzuner, 2015), clinical memorization remains an area of uncertainty in FL.

**Fine-tuning Datasets.** In order to reproduce a plausible FL environment with non-IID data, we select 3 popular medical datasets with different types of QA.

1. *MedMCQA* (Pal et al., 2022) is composed of multiple-choice questions, containing almost 190k entrance exam questions (AIIMS & NEET PG). We fine-tune on the training split and leave aside validation data as a downstream evaluation benchmark.

2. *PubMedQA* (Jin et al., 2019) consists of Yes/No/Maybe questions created from PubMed abstracts. The dataset contains 1k expert-annotated (PQA-L) and 211k artificially generated QA instances (PQA-A). We include 500 questions from the train and validation sets of PQA-L and 50k questions of PQA-A.

3. *Medical Meadow flashcards* (Han et al., 2023) contains 39k questions created from Anki Medical Curriculum flashcards compiled by medical students. We include 10k instances for fine-tuning data.

**Medical Benchmarks.** To measure the downstream performance of the fine-tuned models, we evaluate models on 4 medical benchmarks following existing methodology (Wu et al., 2023a; Singhal et al., 2023b;a; Chen et al., 2023): MedQA, PubMedQA, MedMCQA, and MMLU-Medical.

1. *MedQA's 4-option questions.* MedQA (Jin et al., 2020) consists of US Medical License Exam (USMLE) multiple-choice questions. The test set contains 1278 questions with both 4 and 5-option questions. Following Chen et al. (2023), we report each case separately, respectively MedQA-4 and MedQA.

2. *MedQA's 5-option questions.*

3. *PubMedQA's* test set contains 500 expert-annotated questions. No artificially-generated questions are used during evaluation.

4. *MedMCQA's* test set does not provide answer labels, therefore we rely on the validation set, containing 4183 instances, to benchmark downstream performance following Wu et al. (2023a) and Chen et al. (2023).

5. *MMLU-Medical.* MMLU (Hendrycks et al., 2021) is a collection of 4-option multiple-choice exam questions covering 57 subjects. We follow Chen et al. (2023) and select a subset of 9 subjects that are most relevant to medical and clinical knowledge: high school biology, college biology, college medicine, professional medicine, medical genetics, virology, clinical knowledge, nutrition, and anatomy, and group them into one medical-related benchmark: MMLU-Medical.

We use 3-shot in-context learning without any chain-of-thought reasoning and average the accuracy over 3 seeds.

**Models.** To account for the effect of model size on memorization (Carlini et al., 2023; Tirumala et al., 2022), we study pre-trained models ranging from 1B to 8B parameters: Llama 3.2 1B, Llama 3.2

3B, Llama 3 8B (Dubey et al., 2024), Llama 2 7B (Touvron et al., 2023), and Mistral 7B v0.3 (Jiang et al., 2023).

## 4.2 QUANTIFYING MEMORIZATION

How we measure memorization is largely inspired by Carlini et al. (2023). In short, we inject sensitive sequences, so-called "canaries" (Carlini et al., 2019; Jagielski et al., 2023; Thakkar et al., 2020), into fine-tuning data and then measure the models' ability to regurgitate this information when prompted with the beginning of these sequences. In Appendix C.2, we give an example of memorization scores for Llama 2 7B.

**Canaries.** Unlike prior works that evaluate memorization of all training data (Carlini et al., 2023; Ippolito et al., 2023; Hans et al., 2024), we are interested in measuring how much sensitive information is memorized. Similar to Lehman et al. (2021) and Mireshghallah et al. (2022), we inject medical records into our training set originating from the 2014 i2b2/UTHealth corpus dataset (Stubbs & Özlem Uzuner, 2015). The i2b2 dataset contains 1304 longitudinal medical records that describe 296 patients.

Since data duplication has been shown to greatly influence memorization (Carlini et al., 2023; Lee et al., 2022; Kandpal et al., 2022), we randomly select 30% of the medical records and duplicate them 10 times within our fine-tuning data in order to study data duplication in our experiments.

**Prompting.** To measure unintended memorization after fine-tuning, we randomly select test sequences from the medical records (one sequence per record) and split each sequence into a prefix $p$ and a suffix $s$. Conditioned on the prefix, the model generates text via greedy decoding and the generated suffix is compared with the ground truth. We set the length of the generated suffix $s$ to 50 tokens, in line with Carlini et al. (2023), Ippolito et al. (2023) and Hans et al. (2024).

Following Carlini et al. (2023), we measure the effect of the context size by prompting the model on each test sequence several times with prompts of lengths in $\{10, 50, 100, 200, 500\}$. The different prompts for one test sequence are constructed such that the suffix $s$ is kept identical while varying the prompt length. This ensures a fair comparison between prompt lengths, since different suffixes may be more or less difficult to regurgitate.

**Memorization scores.** To compare generated text with the ground truth, we rely on two metrics: (1) the **exact token match rate** and (2) the **BLEU score** to measure approximate reproduction, as prior works suggest that the exact match rate does not capture subtler forms of memorization (Ippolito et al., 2023). In line with this work, we consider a sequence memorized if the generated suffix and the ground truth yields a BLEU score $> 0.75$. For both metrics, lower is better and a score of 1 denotes the complete memorization of all test sequences. In Appendix C.2, we provide an example for Llama 2 7B fine-tuning.

## 4.3 CENTRALIZED LEARNING

To the best of our knowledge, the impact of LoRA on memorization has not been previously quantified; therefore, we begin by studying LoRA in the context of centralized learning (CL) before considering federated learning (FL).

**Training details.** In the centralized learning setting, we merge *PubMedQA*, *MedMCQA* and *Medical Meadow Flashcards* into one fine-tuning dataset in which we inject the *i2b2* medical records to benchmark memorization after fine-tuning. We use a validation split of 10% and for each model we search for the learning rate yielding the lowest validation loss. More details on hyperparameters can be found in Appendix B.1.

**Accuracy.** To study how LoRA mitigates unintended memorization, we must first assess if it comes at a cost in model performance. Figure 6 illustrates the average accuracy over fine-tuning strategies. Comparing full fine-tuning against LoRA, we find that LoRA comes with a relatively negligible cost in accuracy. Every fine-tuning yields a significant accuracy improvement of the pre-trained model except for Llama 3.1 8B, in which performance minimally improved. We hypothesize that part or all of our fine-tuning dataset has already been trained on during Llama 3.1 8B's pre-training phase. Accordingly, we exclude Llama 3.1 8B from subsequent experiments.

**Memorization.** Given that LoRA matches full fine-tuning performance in our experiments, we now measure the unintended memorization occurring during fine-tuning, illustrated in Figure 1. To account for prompt length, we include a figure (plots **(c)** and **(f)**) for each metric with the highest memorization score obtained across settings, which is systematically reached on duplicated documents with the longest prompt.

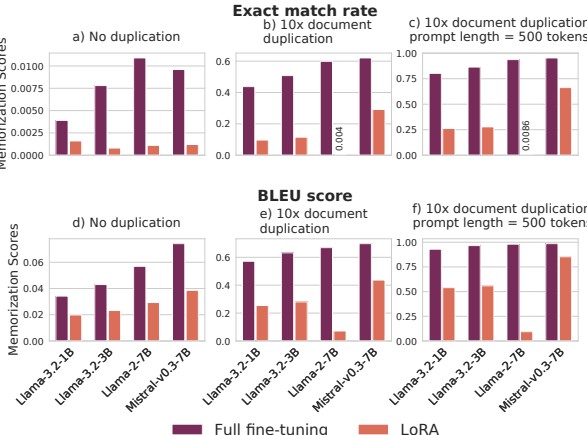

Figure 1: **LoRA vs full fine-tuning memorization scores in centralized learning.** LoRA consistently yields lower memorization scores (lower is better). Unless stated otherwise, scores are averaged across prompt lengths. Values are shown when bars are too small. Right-most figures denote the worst-case setting where memorization scores are the highest. Plots **(a)**-**(c)** show memorization using exact match rate with no duplication, 10x document duplication, and 10x document duplication with a 500 tokens prompt length, while **(d)**-**(f)** use BLEU score.

**Analysis.** Across all model sizes, data duplication greatly increases memorization and longer prompt lengths increase the extraction success. Figure 1 also illustrates that larger models memorize more (Carlini et al., 2023; Tirumala et al., 2022). Most importantly, we see that *models fine-tuned in centralized learning with LoRA consistently exhibit lower memorization scores*, suggesting the adequacy of using of LoRA as a memorization-mitigating technique with little to no performance cost.

Additionally, we compute the memorization scores of pre-trained models without fine-tuning, to obtain control values. This is equivalent to computing the models' ability to "guess" the suffix without having seen previously the medical records. We obtained scores an order of magnitude lower than any fine-tuned model score, which additionally confirms that none of the models had already been trained on the i2b2 dataset. Thus, while some scores in Figure 1 may appear low at first glance, the lowest memorization depicted in this figure is >10 times higher than the control.

### 4.3.1 UTILITY-PRIVACY TRADEOFF

To further confirm that the privacy gains observed on models trained with LoRA do not come at the cost of utility, and that the privacy loss observed with full fine-tuning is not due to overfitting or preventable by early stopping, we analyzed the utility-privacy tradeoff throughout the fine-tuning process. Figure 2 illustrates the evolution of privacy and utility for Llama 3.2 3B during both LoRA and full fine-tuning. The figure shows that LoRA fine-tuning consistently follows a more privacy-preserving trend, with lower memorization scores compared to full fine-tuning at similar utility levels. Furthermore, after a certain number of fine-tuning steps, the model's tendency to memorize data increases without significant improvements in utility, due to overfitting. This highlights that *early stopping during LLM training not only improves efficiency, but also helps privacy by reducing the risk of memorization.*

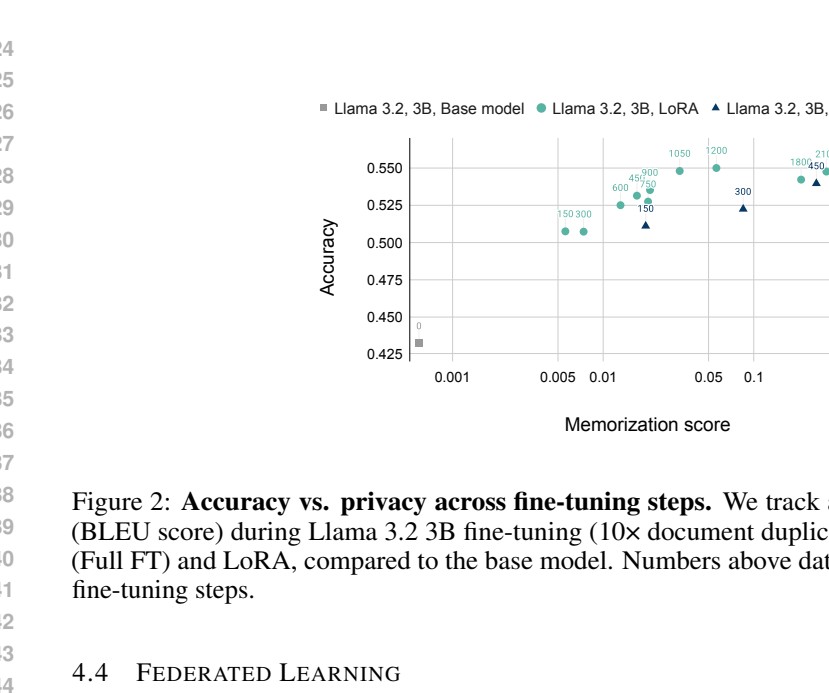

Figure 2: **Accuracy vs. privacy across fine-tuning steps.** We track accuracy and memorization (BLEU score) during Llama 3.2 3B fine-tuning (10× document duplication) using full fine-tuning (Full FT) and LoRA, compared to the base model. Numbers above data points indicate completed fine-tuning steps.

## 4.4 FEDERATED LEARNING

Having empirically measured how LoRA reduces unintended memorization in centralized learning, we now turn to federated learning. The federated learning framework contains multiple key differences with centralized learning that may impact memorization, such as Federated Averaging or non-IID data across participants (Thakkar et al., 2020).

**Training details.** We define a heterogeneous setting with one client per dataset. In other words, we fine-tune models with 3 participants, where each participant trains locally on one of the 3 datasets MedMCQA, PubMedQA, and Medical Meadow flashcards. We split and inject i2b2 medical records into each dataset proportionally to their size. Participants fine-tune over their local dataset for one epoch between each global weight update, for a total of 5 rounds. For every model, we fine-tune the learning rate on each local dataset. More training details are included in Appendix B.

To provide fair comparisons between multiple federated learning fine-tuning, Figures 3 and 5 report metrics for the last federated communication round. This ensures that each model has been fine-tuned on the medical records the same number of times. Additionally, we include the accuracy and memorization metrics for each round in Appendix C.1.

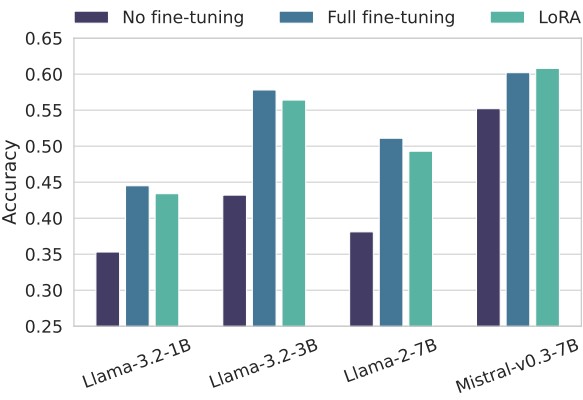

Figure 3: **Downstream accuracy in federated learning.** LoRA yields relatively similar accuracy to full fine-tuning for several LLMs in a heterogeneous FL setting.

**Accuracy.** Figure 3 depicts downstream accuracy of federated fine-tuning. All fine-tunings show relatively similar accuracy values between full fine-tuning and LoRA. This suggests that LoRA is a

competitive technique in federated learning and can replace full fine-tuning at relatively little cost, in addition to lowering the hardware requirements and the communication overheads.

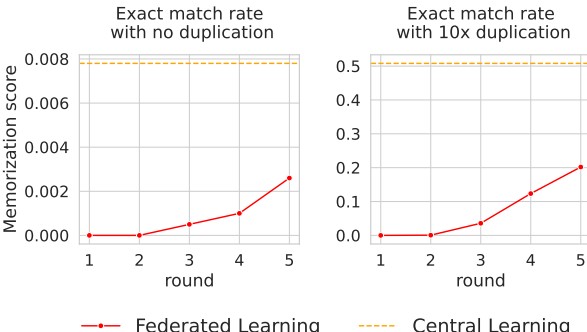

Figure 4: **Exact match rates of FL and CL.** We compare memorization between CL and FL when fine-tuning Llama 3.2 3B.

**Memorization.** We first start by comparing memorization in federated learning to centralized learning in Figure 4. We observe that FL can enhance privacy by reducing memorization. This is consistent with previous work (Thakkar et al., 2020) suggesting that FedAvg and a non-IID data distribution contribute to reducing unintended memorization. However, we note that memorization increases monotonically with the number of rounds (i.e. the number of times medical records are seen). Therefore, a model fine-tuned via FL can reach similar or even greater memorization levels as the number of rounds increases. In fact, Figure 8 shows that, after a certain number of rounds, fine-tuning Llama 2 7B exhibits more memorization across several metrics in FL than in CL. Thus, our results expand on previous work by focusing on how memorization increases throughout the rounds. Comparisons for all models and metrics are included in Appendix C.3.

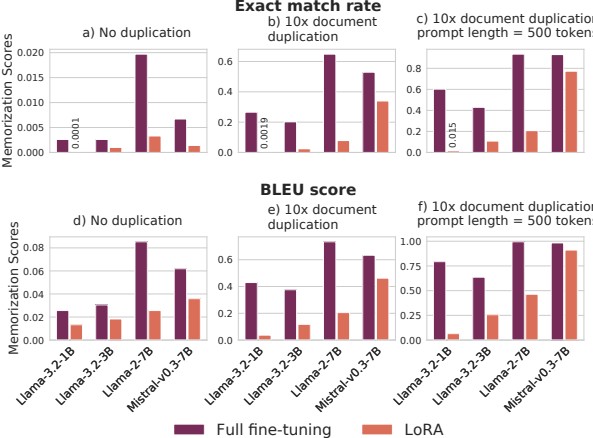

Figure 5: **Memorization of LoRA vs full fine-tuning in federated learning.** LoRA yields significantly lower memorization scores in every setting for an equivalent performance. Plots **(a)**-**(c)** show memorization using exact match rate with no duplication, 10x document duplication, and 10x document duplication with a 500 tokens prompt length, while **(d)**-**(f)** use BLEU score.

**Analysis.** Despite FL showing lower memorization than CL, all federated fine-tunings exhibit significant memorization, thus showing the need for additional privacy-preserving techniques. Figure 5 shows how using LoRA instead of full fine-tuning impacts memorization. *Fine-tuning federated LLMs with LoRA displays lower memorization than full fine-tuning across all metrics and models.* LoRA fine-tuning can reduce memorization up to $10\times$ for a negligible accuracy loss. We do note that the memorization impact of LoRA differs between similarly sized models. For example, fine-tuning

Llama 2 7B with LoRA shows a drastic memorization improvement over full fine-tuning, whereas Mistral v0.3 7B shows a lower impact.

We also find that not all trends observed in centralized learning hold in federated learning: data duplication, longer context and considering paraphrasing all yield higher memorization scores, however Figure 5 shows that bigger models do not necessarily result in more memorization with full fine-tuning, as Llama 3.2 1B reaches higher memorization scores than Llama 3.2 3B. Yet the trend still holds when looking at LoRA fine-tuning. We leave further exploration of how model size influences memorization in federated learning for future work.

Finally, LoRA drastically reduces FL communication overhead. For instance, each round of our setting requires a total data exchange of 74GB for a 7B model, and *using LoRA reduces the load by a factor of 152, decreasing the overhead to 498MB.*

### 4.4.1 SECURE AGGREGATIONS

FL's privacy benefits can be compromised if participants gain access to each other's fine-tuned local models. While Figure 8 highlights reduced memorization after model aggregation, unsecured local models may still expose additional information regarding participants' datasets. In Appendix D, we show how secure aggregation addresses this vulnerability by using a third party to aggregate encrypted local contributions using Fully Homomorphic Encryption (FHE) and decrypting the aggregated model collectively through Secure Multiparty Computation (SMPC), as described in Sébert et al. (2022). Experiments were conducted using the open-source Lattigo library (Lattigo v6; Mouchet et al., 2020).

### 4.5 COMBINING LoRA WITH OTHER METHODS

Although LoRA mitigates unintended memorization on its own, we investigate whether it can be combined with other privacy-persevering techniques without compromising performance or increasing memorization. If users are focused on reducing extractable memorization in pre-training, then they may be interested in Goldfish loss (LoRA is preferred for fine-tuning), but we investigate and verify its potential for fine-tuning. Gradient noising and clipping can be used to satisfy $(\epsilon, \delta)$-differential-privacy guarantees (see Appendix G), which LoRA alone has not been formally proven to provide.

Nonetheless, we emphasize that Goldfish loss and DP noising/clipping are not *efficient* strategies, as both require calculation of the full gradient. Hence, users will choose LoRA if they are concerned about backpropagation costs or communication overhead, which is a common scenario in FL.

### 4.5.1 GOLDFISH LOSS

The Goldfish loss (Hans et al., 2024) has been introduced recently as a memorization mitigating technique for pre-training language models via a new next-token training objective. The training procedure randomly excludes tokens from the loss computation in order to prevent verbatim reproduction of training sequences. In Appendix E, we evaluate the memorization and accuracy of Llama 3.2 3B fine-tuned with LoRA in combination with Goldfish loss. We also compare it to the same model fully fine-tuned with Goldfish loss only. *The combination of LoRA with Goldfish loss synergistically achieves lower memorization beyond what either strategy achieves alone.*

## 5 CONCLUSION AND LIMITATIONS

In this work, we demonstrate that LoRA is capable of reducing memorization of fine-tuning training data. In particular, this effect is observable in both centralized learning and federated learning (FL), and we find this effect is especially pronounced in the latter. Moreover, it is possible to further reduce memorization by combining LoRA with other strategies such as Goldfish loss or conventional privacy-preserving mechanisms such as Gaussian noising and gradient clipping. FL was previously shown to reduce memorization for simple LSTM-based next-word predictors (Hard et al., 2018; Thakkar et al., 2020) and we demonstrate that generative LLMs inherit this benefit as well. However, further theoretical analysis of this phenomenon, which may relate to the LoRA reductive effect, is needed.

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

## A    FURTHER RELATED WORK

**Membership inference attacks** (MIA) rely on rigorous statistical principles to assess privacy risks in machine learning models. (Shokri et al., 2017) introduced an approach for determining whether a specific data point was part of a model's training dataset. These attacks exploit differences in model behavior on training versus non-training data, posing significant privacy concerns for sensitive information. Building on this, (Hongyan et al., 2024) extended these concepts to LLMs by incorporating contextual information. This study demonstrated that LLMs are particularly vulnerable to membership inference attacks, as they often retain verbatim information from their training datasets. The work highlighted the increased privacy risks associated with LLMs due to their scale and training dynamics.

**Secure Aggregations.** While the conventional FL ensures that raw data is not shared between participants during collective training, it does not address the risk of data leakage through model updates shared prior to aggregation. For example, in the honest-but-curious scenario, a server examines whether client data can be reconstructed (Huang et al., 2021). This vulnerability becomes particularly critical with LLMs, given their propensity for memorization. To address the privacy risks associated with local model exchanges in FL, (Truex et al., 2019) proposes a hybrid approach that combines differential privacy with secure multiparty computation (SMC). In this framework, local models are encrypted and remain hidden from other participants prior to aggregation, thereby mitigating privacy leakage risks associated with individual local models by focusing them on the aggregated model during each aggregation round. While this method has been explored for general machine learning applications, to the best of our knowledge, it has not yet been investigated in the context of large language models (LLMs).

## B    TRAINING DETAILS

### B.1    HYPERPARAMETERS

In centralized learning, we sweep the learning rate $\in \{1e - 5, 5e - 5, 1e - 4, 5e - 4\}$ for full fine-tuning experiments. For LoRA experiments, we search for learning rate values $\in \{5e - 5, 1e - 4, 5e - 4, 1e - 3\}$. In federated learning experiments, we sweep the learning rate on each dataset individually for one epoch, with the same set of values as in centralized learning.

For all experiments we fine-tune models with the AdamW optimizer (Loshchilov & Hutter, 2019) with default parameters ($\beta_1 = 0.9$, $\beta_2 = 0.999$, $\epsilon = 1e^{-8}$, weight decay of 0.01). We used a context length of 1024 and ensured that no text inputs were longer than the context length. We use a linear warmup of 100 steps with a cosine annealing schedule. Unless mentioned otherwise, we use a global batch size of 32 with gradient accumulation and gradient checkpointing. For all LoRA experiments with use a rank of 16, an alpha of 8, drop out 0.05 and use adapters for all projection layers. Additionally, we study the impact of the LoRA rank on memorization in Section B.2.

### B.2    THE LORA RANK AND MEMORIZATION

We measure the influence of the LoRA hyperparameters by varying the rank and measuring the resulting memorization. We study rank values $r \in \{4, 16, 64, 128, 256, 1024\}$ and set alpha to twice the rank, following common practice. We decrease the learning rate exponentially as the rank increase.

As shown in Table 1, increasing the rank, i.e. increasing the number of weights updated during fine-tuning, results in more memorization, ranging from virtually no verbatim memorization with a rank of 4 to almost 50% of the medical records being memorized for rank 1024 when considering duplicated medical records. We note that in our case, larger ranks do not necessarily imply better accuracy. We hypothesize that larger ranks might make overfitting more likely to occur. Additionally, each rank value can benefit from more extensive hyperparameter tuning.

Table 1: **Impact of the LoRA rank on memorization.** We fine-tune Llama 3.2 3B with LoRA in centralized learning on increasing LoRA ranks. We find that higher ranks lead to more memorization.

| LoRA rank | Exact match rate | | BLEU Score | | Accuracy |
|---|---|---|---|---|---|
| | No duplication | 10x duplication | No duplication | 10x duplication | |
| 4 | 0.0003 | 0 | 0.0133 | 0.0198 | 0.509 |
| 16 | 0.0005 | 0.0031 | 0.0167 | 0.0623 | 0.512 |
| 64 | 0.0031 | 0.2105 | 0.0258 | 0.379 | 0.511 |
| 128 | 0.0042 | 0.3735 | 0.0305 | 0.5111 | 0.510 |
| 256 | 0.0057 | 0.4895 | 0.0352 | 0.5809 | 0.542 |
| 1024 | 0.0063 | 0.4981 | 0.0409 | 0.6228 | 0.530 |

# C  AUXILIARY RESULTS

## C.1  ACCURACY

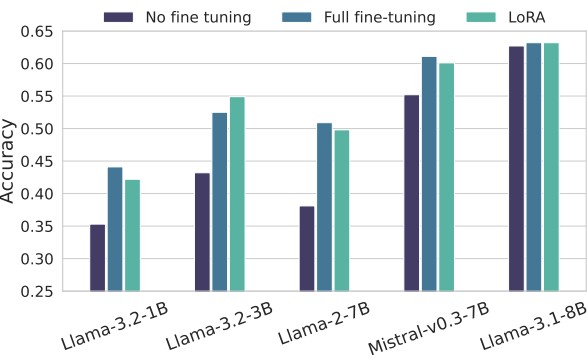

Figure 6: **Downstream accuracy of centralized learning averaged across the 5 benchmarks.** LoRA matches full fine-tuning accuracy on every model tested. We report the out-of-the-box accuracy of the pre-trained models as a control. A breakdown per benchmark is included in Table 2.

Table 2 includes a breakdown per benchmark of the downstream accuracy of LoRA and full model fine-tuning in centralized learning as well as performance of pre-trained models without fine-tuning. Table 3 shows the accuracy of federated fine-tuning per round.

Table 2: **Downstream accuracy in central learning.** Best accuracy values are marked in **bold**.

| Model | Fine-tuning | MMLU-medical | PubMedQA | MedMCQA | MedQA | MedQA-4 | Average |
|---|---|---|---|---|---|---|---|
| Llama 3.2 1B | No fine-tuning | 0.353 | 0.363 | 0.49 | **0.329** | 0.275 | 0.308 |
| | Full | **0.456** | **0.616** | **0.431** | 0.322 | **0.379** | **0.441** |
| | LoRA | 0.447 | 0.594 | 0.397 | 0.312 | 0.362 | 0.422 |
| Llama 3.2 3B | No fine-tuning | 0.432 | 0.597 | 0.122 | 0.491 | 0.446 | 0.504 |
| | Full | 0.59 | 0.536 | **0.542** | **0.452** | **0.507** | 0.525 |
| | LoRA | **0.608** | **0.676** | 0.512 | 0.448 | 0.5 | **0.549** |
| Llama 2 7B | No fine-tuning | 0.381 | 0.426 | 0.452 | 0.380 | 0.292 | 0.353 |
| | Full | **0.562** | 0.596 | **0.516** | **0.395** | **0.478** | **0.509** |
| | LoRA | 0.560 | **0.726** | 0.448 | 0.353 | 0.405 | 0.498 |
| Mistral v0.3 7B | No fine-tuning | 0.552 | 0.635 | 0.7 | 0.483 | 0.438 | 0.503 |
| | Full | 0.659 | **0.758** | **0.588** | **0.499** | **0.551** | **0.611** |
| | LoRA | **0.667** | **0.758** | 0.572 | 0.467 | 0.54 | 0.601 |

Table 3: **Downstream accuracy per federated round**. We emphasize in **bold** the earliest round where models reach their best accuracy.

| Model | Fine-tuning | Accuracy per round | | | | |
|---|---|---|---|---|---|---|
| | | 1 | 2 | 3 | 4 | 5 |
| Llama 3.2 1B | Full | 0.425 | 0.438 | 0.444 | **0.445** | 0.445 |
| | LoRA | 0.415 | 0.422 | 0.430 | 0.432 | **0.434** |
| Llama 3.2 3B | Full | 0.541 | 0.561 | 0.554 | 0.573 | **0.578** |
| | LoRA | 0.557 | **0.564** | 0.559 | 0.563 | 0.564 |
| Llama 2 7B | Full | 0.468 | 0.488 | 0.482 | 0.495 | **0.511** |
| | LoRA | 0.475 | 0.490 | 0.482 | **0.494** | 0.493 |
| Mistral v0.3 7B | Full | 0.181 | 0.590 | 0.599 | **0.603** | 0.602 |
| | LoRA | 0.594 | 0.599 | 0.598 | 0.604 | **0.608** |

C.2  MEMORIZATION SCORE

Figure 7 illustrates with Llama 2 7B multiple trends that are consistent with results previously mentioned:

1. *There is significantly, and alarmingly, more memorization when the medical records occur multiple times in the fine-tuning data.*

2. *Longer prompts show higher memorization (discoverability phenomenon).*

3. *There is significantly more memorization with approximate generation (BLEU score).*

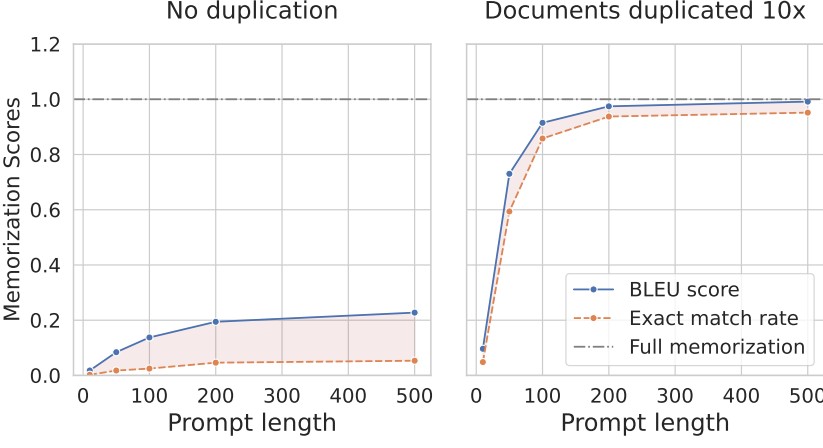

Figure 7: **An example of memorization scores for a full fine-tuning of Llama 2 7B.** We report the exact match rate and BLEU score with respect to the prompt length, with and without duplication. We also show the memorization upper bound ("Full memorization") reached when every test sequence has been memorized.

C.3  MEMORIZATION SCORES IN FL

Figure 8 shows the memorization scores per round of federated learning. We can see that using LoRA results in lower unintended memorization than full fine-tuning at every round.

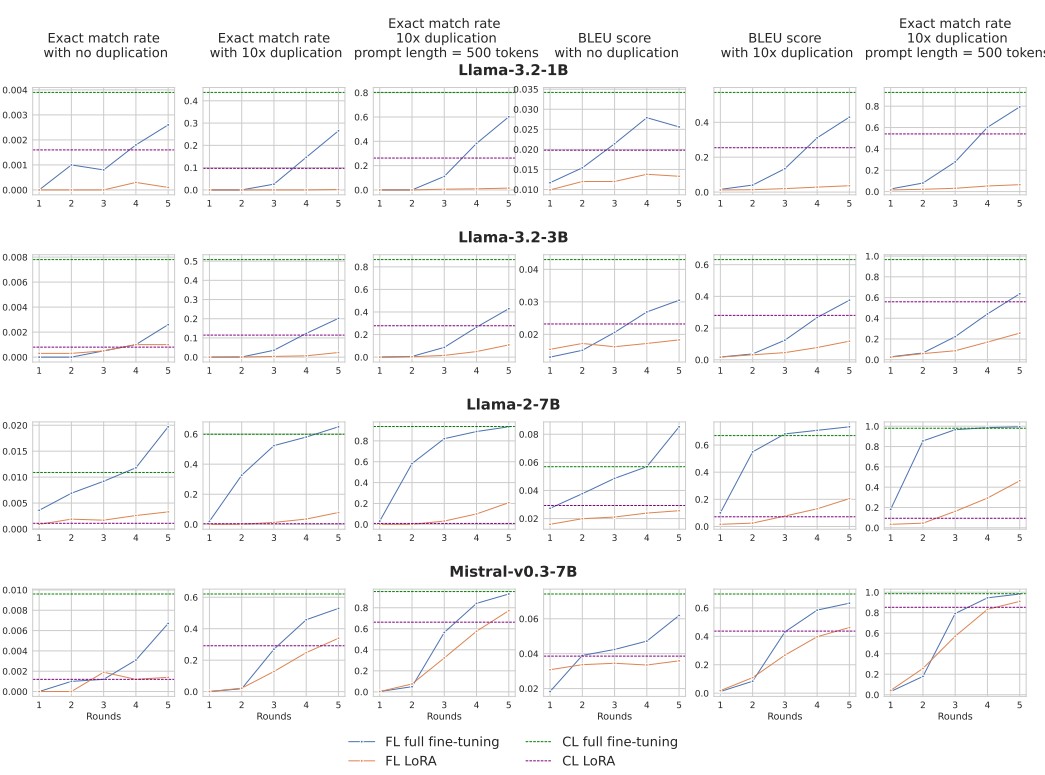

Figure 8: **Memorization scores for central learning and federated learning with respect to rounds.** In all settings, LoRA results in better privacy than a full fine-tuning.

## D SECURE AGGREGATIONS

Secure aggregations ensure that sensitive data remains protected and prevents the aggregator from decrypting any model. We evaluate the runtime performance of using secure aggregation in conjunction with LoRA in an FL setting.

**Performance.** To evaluate the performance impact of secure aggregation, we use Lattigo, an open-source library that enables secure protocols based on multiparty homomorphic encryption Lattigo v6; Mouchet et al. (2020). Specifically, it implements the CKKS scheme, which allows efficient encrypted computations on real-valued data, making it ideal for the secure aggregation of the LoRA models trained by the clients/participants. In our experiments, we consider 3 clients and configure CKKS parameters to enable 32-bit precision. Since our LoRA models are trained with 16-bit precision, this ensures that **secure aggregation does not introduce any accuracy loss** compared to standard aggregation in plaintext.

Secure aggregation introduces a time overhead due to encryption, homomorphic operations, and collective decryption. The duration of encrypted aggregation is influenced by the number of weights being aggregated, specifically the number of LoRA weights. In our experiments with Llama 3.2 3B, **a LoRA update contains 24,772,608 parameters, representing approximately 0.77% of the full model's parameters**. In Table 4, we report the aggregation times for vectors of varying sizes, corresponding to the number of LoRA weights. Aggregating three vectors of the size of our LoRA takes 11.33 seconds, which is negligible compared to the time required for local fine-tuning at each round.

Table 4: **Execution Time of the Secure Aggregation Protocol.** The protocol aggregates three equal-sized encrypted vectors for varying sizes.

| Aggregation Length | Time Taken |
|---|---|
| $10^1$ | 12.16ms |
| $10^2$ | 11.61ms |
| $10^3$ | 11.32ms |
| $10^4$ | 17.29ms |
| $10^5$ | 58.91ms |
| $10^6$ | 474.46ms |
| $10^7$ | 4.37s |
| $2.48 \times 10^7$ (LoRA size) | **11.33s** |
| $10^8$ | 68.24s |

## E  GOLDFISH LOSS

In this section, we evaluate how LoRA combined with Goldfish loss impact the accuracy and the memorization of Llama 3.2 3B. While Goldfish loss has been designed for pre-training, we apply it to our fine-tuning and report values for various dropping frequencies $k$. We use a hashing context width $h = 13$ following the authors' methodology (Hans et al., 2024).

Table 5 shows how combining Goldfish loss with LoRA mitigates memorization compared to a full fine-tuning. By contrasting memorization scores with control values, we can also note that the Goldfish loss is an effective memorization-mitigation technique.

Table 5: **Impact of Goldfish loss on BLEU Scores and accuracy in LoRA Fine-Tuning.** Llama 3.2 3B is fine-tuned with different dropping frequencies ($k$). Best accuracy is marked in **bold**.

| Goldfish $k$ | BLEU, no duplication | BLEU, 10x duplication | Accuracy |
|---|---|---|---|
| 2 | 0.0133 | 0.0216 | 0.514 |
| 3 | 0.0154 | 0.0426 | **0.549** |
| 4 | 0.0180 | 0.0543 | 0.534 |
| 5 | 0.0183 | 0.0815 | 0.540 |
| 10 | 0.0256 | 0.1494 | 0.538 |
| 100 | 0.0266 | 0.2852 | 0.537 |
| 1000 | 0.0256 | 0.3111 | 0.533 |
| 10000 | 0.0253 | 0.2944 | 0.545 |
| Control | 0.0245 | 0.2920 | 0.550 |

To assess the impact of LoRA in combination with Goldfish loss, we evaluated the memorization and accuracy of fine-tuning the same model using full fine-tuning. Table 6 presents the memorization scores and accuracy of the model fine-tuned with Goldfish loss alone, without LoRA. Our results indicate that while Goldfish loss reduces memorization, it does not achieve the same level of reduction as the combination with LoRA, especially when duplication occurs in the fine-tuning data. In summary, combining LoRA with Goldfish loss allows a privacy-utility tradeoff that cannot be achieved using Goldfish loss alone.

Table 6: **Impact of Goldfish loss on BLEU Scores and accuracy.** The BLEU scores and the accuracy of Llama 3.2 3B is reported for full fine-tuning across different dropping frequencies ($k$). Best accuracy is marked in **bold**.

| Goldfish $k$ | BLEU, no duplication | BLEU, 10x duplication | Accuracy |
|---|---|---|---|
| 2 | 0.0146 | 0.0340 | 0.517 |
| 3 | 0.0243 | 0.0679 | 0.513 |
| 4 | 0.0282 | 0.1148 | 0.524 |
| 5 | 0.0310 | 0.1568 | 0.521 |
| 10 | 0.0342 | 0.3006 | **0.545** |
| 100 | 0.0399 | 0.5821 | 0.534 |
| 1000 | 0.0425 | 0.6235 | 0.527 |
| 10000 | 0.0407 | 0.6235 | 0.516 |
| Control | 0.0417 | 0.6235 | 0.538 |

# F  NEFTune

NEFTune is a regularization technique consisting in adding random noise to the embedding vectors to improve instruction fine-tuning. While not introduced as a privacy-preserving technique per se, we hypothesize that a fine-tuning regularization such as NEFTune may also reduce unintended memorization.

We display results after applying NEFTune with noise value $\alpha \in \{5, 10, 15, 30, 45\}$. We find that adding noise does not improve accuracy when applied to our domain adaptation fine-tuning. Secondly, increasing the noise does not yield better privacy, at least not until we set alpha to 45, which is greater than alpha values reported by the original work (5, 10, and 15).

Table 7: **NEFTune impact on the BLEU score and accuracy when combined with LoRA.** We analyze LoRA fine-tuning with Llama 3.2 3B and different noise scaling factors $\alpha$.

| $\alpha$ | No duplication | 10x duplication | Accuracy |
|---------|---------------|-----------------|----------|
| Control | 0.0276 | 0.4170 | 0.562 |
| 5 | 0.0284 | 0.4525 | 0.560 |
| 10 | 0.0300 | 0.4506 | 0.518 |
| 15 | 0.0284 | 0.4525 | 0.544 |
| 30 | 0.0282 | 0.4377 | 0.548 |
| 45 | 0.0248 | 0.3599 | 0.518 |
| 60 | 0.0227 | 0.2759 | 0.501 |
| 100 | 0.0183 | 0.1006 | 0.391 |

# G  DIFFERENTIAL PRIVACY

$(\epsilon, \delta)$-Differential privacy (DP) provides formal guarantees that an individual's data cannot be inferred from a model's output, by quantifying the model's sensitivity to changes in input data. Following Li et al. (2021) and Liu et al. (2024), we define sensitivity as the maximum change in model output resulting from the inclusion or removal of a single data point in the training dataset (record-level DP).

Implementing DP requires modifications to the fine-tuning pipeline to limit the influence of individual data points on model parameters. Gradient clipping, which constrains the magnitude of gradient updates, is a key technique in this process. In our experiments (see Appendix G.1), applying a gradient clipping value of 0.0001 significantly reduces memorization and improves accuracy compared to the default value of 1.0. This demonstrates gradient clipping as a privacy-enhancing method in itself, even without the addition of noise. But the use of stochastic gradient descent (SGD), required for DP-SGD, presents challenges in fine-tuning the Llama 3.2 3B model. Despite an extensive search for optimal learning rates, SGD consistently underperforms compared to Adam-derived optimizers (see Appendix G.2).

## G.1  GRADIENT CLIPPING

Table 8 illustrates the effect of different gradient clipping values on the BLEU score and accuracy achieved during the fine-tuning of LLama 3.2 3B.

## G.2  OPTIMIZER EFFECT ON LOSS

Figure 9 illustrates the loss reduction difference between Stochastic Gradient Descent (SGD) and Paged AdamW optimizers during the fine-tuning of Llama 3.2 3B. The SGD optimizer failed to achieve the same level of loss reduction as Paged AdamW.

# H  POST-FINE-TUNING GAUSSIAN NOISE INJECTION

This section provides details and results of the injection of noise into the weights of a model after fine-tuning. Specifically, the noise is sampled from a Gaussian distribution $\mathcal{N}(\mu, \sigma^2)$, where the mean $\mu$ is set to 0, and $\sigma^2$ is the variance that determines the noise's magnitude. Unlike the DP

Table 8: **Gradient clipping impact on the BLEU score and accuracy.** The BLEU score and the accuracy of Llama 3.2 3B is reported for LoRA fine-tuning. Best accuracy is marked in **bold**.

| Clipping Value | No duplication | 10x duplication | Accuracy |
|---|---|---|---|
| $1.0 \times 10^{0}$ (default) | 0.0266 | 0.4235 | 0.520 |
| $5.0 \times 10^{-1}$ | 0.0235 | 0.4235 | **0.541** |
| $1.0 \times 10^{-1}$ | 0.0229 | 0.4031 | 0.530 |
| $5.0 \times 10^{-2}$ | 0.0243 | 0.3827 | 0.534 |
| $1.0 \times 10^{-2}$ | 0.0227 | 0.3914 | 0.506 |
| $5.0 \times 10^{-3}$ | 0.0245 | 0.3914 | 0.531 |
| $1.0 \times 10^{-3}$ | 0.0250 | 0.3352 | 0.519 |
| $5.0 \times 10^{-4}$ | 0.0203 | 0.2914 | 0.528 |
| $1.0 \times 10^{-4}$ | 0.0185 | 0.0926 | 0.536 |
| $5.0 \times 10^{-5}$ | 0.0151 | 0.0438 | 0.506 |
| $1.0 \times 10^{-5}$ | 0.0086 | 0.0099 | 0.491 |
| $5.0 \times 10^{-6}$ | 0.0065 | 0.0080 | 0.449 |
| $1.0 \times 10^{-6}$ | 0.0026 | 0.0012 | 0.460 |
| $5.0 \times 10^{-7}$ | 0.0026 | 0.0012 | 0.392 |
| $1.0 \times 10^{-7}$ | 0.0026 | 0.0012 | 0.377 |

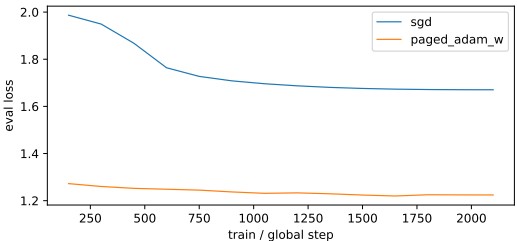

Figure 9: **Loss reduction comparison between optimizers.** The plot compares loss reduction during the fine-tuning of Llama 3.2 3B using different optimizers: SGD (blue) and Paged AdamW (orange).

Gaussian mechanism, this approach does not provide formal privacy guarantees. However, it offers a practical and computationally light method to mitigate the memorization of sensitive information, as it does not require additional fine-tuning and can be directly applied to previously fine-tuned LLMs. Additionally, measuring the performance of this method can illustrate how other noise mechanisms similar to those used in DP might affect accuracy and privacy metrics.

In Table 9, we evaluate its effect under various noise magnitudes, along with the corresponding impact on model accuracy. We applied Gaussian noise to the LoRA weights of a fine-tuned Llama 3.2 3B model, as evaluated in earlier sections. We then compared the model's BLEU score and accuracy across different noise magnitudes.

Table 9: **Impact of noise addition on BLEU score and accuracy.** Llama 3.2 3B is fine-tuned with LoRA across various noise magnitudes ($\sigma$)

| Noise Scale ($\sigma$) | BLEU, no Duplication | BLEU, 10x Duplication | Accuracy |
|---|---|---|---|
| 0 (no noise) | 0.0206 | 0.3012 | 0.553 |
| 0.001 | 0.0211 | 0.3049 | 0.552 |
| 0.01 | 0.0206 | 0.2877 | 0.551 |
| 0.02 | 0.0143 | 0.0994 | 0.541 |
| 0.03 | 0.0083 | 0.0111 | 0.511 |
| 0.04 | 0.0013 | 0.0006 | 0.384 |
| 0.05 | 0.0000 | 0.0000 | 0.110 |

We observe that the accuracy remains unaffected up to a certain noise level ($\sigma = 0.01$) and even shows slight improvement. However, beyond this threshold, accuracy decreases and reduction in memorization similarly follows, appearing to correlate with this decrease. These observations suggest that this mechanism effectively reduces excessive memorization in models that have overfitted onto

their training data. Therefore, this approach offers an alternative to early stopping for controlling memorization which can be applied post fine-tuning. Figure 10 compares the privacy and utility of Llama 3.2 3B subject to post-fine-tuning gaussian noise injection with the evolution of the model fine-tuned with LoRA accross iterations. The noisy model, represented by red dots, has been fine-tuned for 2100 iterations before injecting the gaussian noise. Gaussian noise injection of standard deviations of $\sigma = 0.2$ and $\sigma = 0.3$ have been reported in the plot.

### H.1 PRIVACY-UTILITY TRADEOFF WITH GAUSSIAN NOISE INJECTION

Figure 10 presents a dot plot comparing the privacy-utility tradeoffs of Llama 3.2 3B when fine-tuned with LoRA versus when Gaussian noise is injected after fine-tuning with LoRA. The results indicate that Gaussian noise injection does not enhance the privacy-utility tradeoff compared to fine-tuning with LoRA.

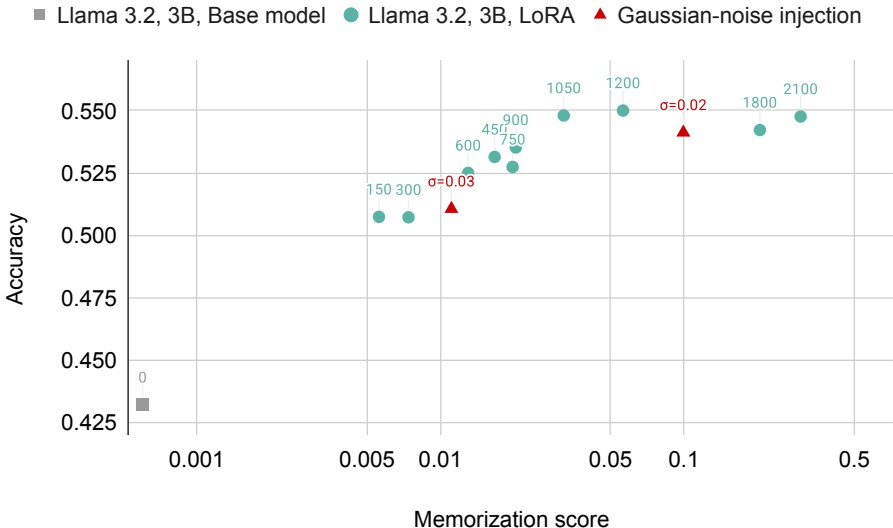

Figure 10: **Privacy-Utility tradeoff with post-fine-tuning gaussian noise injection.** Accuracy and memorization (BLEU score with 10x document duplication) tradeoff of Llama 3.2 3B subject to post-fine-tuning gaussian noise injection with standard deviation. Values above the dots correspond to the number of iterations for LoRA fine-tuning evolution, and the standard deviation of injected noise for noisy models.

