# OpenReview forum: "Mitigating Unintended Memorization with LoRA in Federated Learning for LLMs"
_ICLR.cc/2025/Workshop/BuildingTrust — Submitted to BuildingTrust_

### Official Review · Reviewer_j6op · 2025-02-20
**Technical Limitation exists**

**Rating:** 4
**Confidence:** 3

**Review:**

Experiments primarily focus on Llama 2 and Llama 3 models, lacking validation on other mainstream architectures (e.g., GPT, BERT, PaLM). This raises concerns about whether the findings generalize across diverse model families.

The use of medical records as synthetic “canaries” aligns with the medical application scenario but fails to test memorization of other sensitive data types (e.g., financial records, personal identifiers). This limits the broader applicability of the conclusions.

The privacy-preserving effect of LoRA is not rigorously quantified. Metrics like differential privacy (DP) parameters (ε, δ) or empirical privacy budgets are absent, leaving the actual privacy guarantees unclear.

While the paper observes that LoRA reduces memorization, it does not deeply investigate why this occurs. Simply attributing it to “reduced trainable parameters” is insufficient. Critical unanswered questions include:

Does LoRA’s low-rank update structure inherently limit overfitting to specific training samples?
How do low-rank adaptations alter the model’s representation space to suppress memorization?
Is memorization reduction tied to the rank of LoRA’s update matrices?

---

### Official Review · Reviewer_8b4M · 2025-02-25
**Interesting core finding with limitations in framing.**

**Rating:** 5
**Confidence:** 4

**Review:**

I find the core finding of <<LoRA memorizing less>> interesting, and the results on the medical domain showing that this "gain in privacy" comes at virtually no cost in utility is promising. However, I find the current framing of the paper a bit problematic/too constraining:

- First, there is no mention of Biderman et al. (2024), even though it is a prominent paper showing a very related result on LoRA. This paper can then be viewed as basically an extension to that, explicitly investigating memorization. As Biderman et al. (2024) already showed that LoRA "learns less", framing the memorization results correctly w.r.t. to their findings is also crucial for assessing the novelty of the findings of this paper.
- This leads onto my second concerns on the framing; while I understand that the medical domain makes memorization concerns immediately clear, the experiments could use datasets from other domains as well, and then the framing of the paper could be widened to reach a broader audience.
- Similarly, I do not quite understand the need of committing to Federated Learning in the title and in the intro so early, especially that half of the main experiments revolve around the centralized setting.
- Finally, I find the message that is at some points suggested, i.e., that using LoRA for the benefit of less memorization a bit problematic. While in the experiments of the authors there is no significant utility loss, this is not a general truth, as also shown in Biderman et al. (2024). Therefore, the message has to be more nuanced here.

In general, I think the paper could be framed more generally as  "LoRA memorizes less", leading to potentially a much larger reach and impact, and would benefit from a clear connection to Biderman et al. (2024).

**References**

D Biderman et al., LoRA Learns Less and Forgets Less. TMLR 2024.

---

### Official Review · Reviewer_JFsy · 2025-03-02
**Paper Reivew**

**Rating:** 5
**Confidence:** 4

**Review:**

> Summary:

This paper investigates the unintended memorization problem in FL-trained large language models settings and demonstrates that LoRA can be potentially an effective approach in reducing memorization.

> Pros:

1.The study explores the role of LoRA in mitigating unintended memorization in LLMs, which is an interesting and timely research topic with significant implications for privacy in federated learning and beyond.

2.The paper is well-structured and presents its ideas with clarity, making it easy to follow.

> Cons:

1.One major weakness of the paper is that while it demonstrates LoRA's potential effectiveness in reducing memorization in both CL and FL settings, it does not delve into the underlying reasons or provide potential explanations for this effect, which limits the depth of the study.

2.In Figure 8, the memorization behaviors across different models vary under the same training conditions (full fine-tuning / LoRA training) in both CL and FL settings. However, the authors do not offer an in-depth explanation of these results.

---

### Decision · Program_Chairs · 2025-03-04

**Decision:**

Reject

**Comment:**

This paper explores unintended memorization in FL-trained LLMs and finds that LoRA reduces memorization with minimal utility loss, particularly in the medical domain. However, the framing needs refinement, as it lacks broader validation across models and datasets, does not rigorously quantify privacy gains, and omits connections to prior work demonstrating similar findings.